# Catholic Churches of the Colonial Period in the Southern Andes of Peru: An Evocation towards the Past

Jesús Wiliam Huanca-Arohuanca [1], Edgar Gutiérrez-Gómez [2,*], Giovana Ccama-Ramos [3], Felipe Supo-Condori [3] and Dometila Mamani-Jilaja [3]

[1] Department of Philosophy, Universidad Nacional de San Agustín de Arequipa, Arequipa 04000, Peru; jhuancaar@unsa.edu.pe

[2] Department of Philosophy, Universidad Nacional Autónoma de Huanta, Huanta 05121, Peru

[3] Department of Education, Universidad Nacional del Altiplano de Puno, Puno 21000, Peru; giovanaccamaramos@hotmail.com (G.C.-R.); fsupo@unap.edu.pe (F.S.-C.); domamani@unap.edu.pe (D.M.-J.)

* Correspondence: egutierrez@unah.edu.pe

**Abstract:** The spectral space of the Andes has an architectural cultural richness based on the construction of shrines with a Christian tendency from Spain. The purpose of the study is to analyze and describe the historical process of the colonial Catholic churches located in the Aymara Altiplano of Peru. The study followed the qualitative parameter of historical documentation, where the colonial origin of the buildings was traced, their singular conceptual designs were established, and their use was examined, to finally explain their permanence in time, since they developed in two transcendental moments: the diachronic history and the systematic description of the characteristics in the current churches. Ecclesiastical sanctuaries such as San Pedro and San Pablo, San Juan Evangelista, Concepcion, and Santa Barbara began their construction in 1560, being the first and the oldest to be constituted by the Spanish, but strictly with functions of Christianizing and the collection of taxes from the indigenous people of the Altiplano.

**Keywords:** Catholic Church; Christianization; architecture; history; indigenous people

## 1. Introduction

The Catholic Church has been inserted in Peru from the beginning of the sixteenth century and its duration remains until the present time; its arrival and pretension fractured the different forms of worldview developed in the Andes. Therefore, it is necessary to have in mind three moments: the colonial period that begins in the sixteenth century and lasts until the end of the eighteenth century, the process of independence accompanied by the republic in the nineteenth century, and the republican process of the twentieth and twenty-first century. The colonial period was key to the spread of the Catholic faith among the indigenous population thanks to priests and missionaries (Esquivel 2000). In addition, the Church actively participated in the organization and management of the colonial administration, establishing a network of parishes and dioceses throughout the Peruvian territory. Such is so. The Church also played a role in education and culture during that period. It founded colleges, seminaries, and universities, such as the Universidad Nacional Mayor de San Marcos in Lima, and was responsible for the preservation and dissemination of music, art, and Baroque architecture (Castillo 2019).

After Peru's independence in 1821, the Catholic Church continued to play a relevant role in Peruvian society (Molina Fuentes 2012). However, during the nineteenth century, the Church faced tensions with Republican governments, especially on issues such as education and land ownership. However, throughout the 20th century, the Catholic Church in Peru continued to be an influential institution and sometimes played a leading role in the defense of human rights and social justice. During the 1970s and 1980s, amid political violence, the Church played an active role in defending human rights, the most prominent

role in attempting to atone for its errors during the colony. Despite the fact that, in recent decades, the Catholic Church has continued to work in areas such as education, care for the poor and marginalized, and the promotion of peace and reconciliation, Peru has undergone profound changes; if 30 years ago it had 20 million inhabitants, now this figure exceeds 32 million; therefore, it is assumed that the community of believers in the Catholic religion should increase, however, no such situation has occurred, since, of the 90% of Peruvians who have professed Christianity, Catholics now barely exceed 75% (Sausa 2018). This leads to the following question: What is happening with the Catholic religion in Peru? One of the answers lies in the impact that the Church has had on indigenous people.

Since the arrival of the West to the pristine lands of Latin America (Huanca-Arohuanca and Contreras 2021; Barria-Asenjo et al. 2022), the most representative material emblem of Christian religious expansion has been Catholic temples or churches. In these "colossal buildings are summarized the various methodological resources of the catechization process that the peninsular colonizers used in one of their primary objectives of domination: Christianization" (Mendoza 2016, p. 54). Therefore, in response to the above, at the end of the 20th century, a group of anthropology scholars organized the Symposium on Inter-ethnic Conflicts in South America in Barbados, a study promoted by the Program to Combat Racism of the World Council of Churches and by the Department of Ethnology of the University of Bern (Espinosa 2018), with the purpose of rethinking the Christian Catholic Church and its harmful effects on native peoples. The result of the event "describes and criticizes the role played by state, Churches and anthropology in that history of domination [ . . . ] that accepted the challenge of rethinking their relationship and practices with indigenous peoples" (Espinosa 2018, p. 274).

Undoubtedly, institutions such as the state and Church have promoted a cult of domination and the concealment of the criteria of inequality. In this regard, Mendoza (2016) clarifies that:

> The Church was from its origins a space dedicated to religious worship and a repository of sculptural and pictorial images of the symbols of Christianity. It was also, and continues to be, the center of the preaching of the Gospel, through oratory and sermons, of music and sacred song, as well as of all forms of theatrical staging of religious rites and ceremonies, implanted by the Christian tradition in the European world, who moved to invaded America, with the same fervor and pomp instituted by Christianity from Rome, the fundamental center of the spread of the new religion born in Israel. (Mendoza 2016, p. 54)

Indeed, the Church has inserted itself into the blood and subconscious of the defeated, preaching the illusory truth of appeasing the disasters caused by those who brought iron weapons and the forgiveness that God would have for those who worshiped Pachamama, given that, for the Church, giving thanks to nature is like making a social contract with Satan, an evil that supposedly only existed in Latin America and the natives who inhabited it. Likewise, for its domination to be efficient, the Church needed a material symbol, and this is typified today in the temples built with the pain and blood of thousands of indigenous people. In fact, neither Christ nor Mohammed implanted such a withering ecclesial architectural construction formula against the syncretism and archetype (González 2009) of the indigenous people of the aqu (in Quechua, sand).

On the other hand, and returning to the present, the conditions of space and time turn out to be very important, insofar as that the depth of the changes that history has undergone in the present is evident, whose plot involves the simultaneity of the non-simultaneous (Fazio and Fazio 2018) in the Aymara ontological subjects who live in the present, remembering the afflicted and dark colonial past. Thus, for Topolski (1973) historical time is extremely important, but is also essential to account for factual events in regional history, because historical events must be interpreted in a positive and dialectical manner, interconnecting each event with the maximum of its determinant, listed as a philosophical problem of cultural identity in historical space–time. However, the process of the monumentalization of churches is due to the work of confining the ideology of the indigenous population in

the permanent colonization that comes as a canon of archetype and socialization. Therefore, the analysis of ecclesiastical architecture and historical documentation on the use that the Spanish and indigenous people gave to the temples in terms of political–colonial readjustment is related to settlement, territory, and the emergence of religion (Paredes Cisneros 2018).

From this point, in the vast territory of the Andes region, there are still architectural works from the viceregal era based on the construction of churches, and, currently due to a lack of appraisal, many of them have gradually deteriorated. Taking this reality into account, this manuscript aims to describe the trajectory of the historical past for the sake of the contribution of the certain complexities that make possible the analysis, reflection, and valuation of the historical legacies that are cultural sources in the Andean vertebral matrix. The history of present time allows us to think of a time in which processes were socially and subjectively thought of as current. A history of coetaneity allows us to get closer to those who lived throughout history and are our contemporaries (Montaño 2018).

## 2. Results

The study presents, from the perspective of Frisancho (1999), four churches built in the colonial period, which today are still witnessed with a certain materiality and tremendous belief in the district of Ácora. Such historical monuments are:

(a)    The Church of San Pedro and San Pablo.
(b)    The Church of San Juan Evangelista.
(c)    La Concepción (La Concebida) Church.
(d)    Santa Barbara Church.

For a better understanding of their historical development, it was necessary to study each of these churches in their most particular and peculiar state, considering aspects such as: the start of construction, the construction promoters, the masters in charge of the construction, the completion of the construction, the phases of reconstruction, and the current conditions of each of the churches.

**The Church of San Pedro and San Pablo**
*Home and construction promoters*

During the government of Viceroy Lope García de Castro from 1564 to 1569 (Vargas 1966) and prior to the orders of Viceroy Toledo, Garci Diez de San Miguel visited the province of Chucuito and the neighboring towns located in the circumlacustrine space of Lake Titicaca. The purpose of the visit was to verify the various activities and works carried out at the time, as well as verify the tax population; in this regard, Garci specified in his writings that, in 1567, the construction of a church in Ácora had begun, a town that belonged to the Chucuito district, and the church was built in honor of the dedication of the apostles San Pedro and San Pablo, a name that lasts until the present day.

On the other hand, Gutiérrez et al. (1978), in their research on the architecture of the Altiplano based on church construction, clearly state, in support of the documents found in the La Paz Archive, that the Church of San Pedro and San Pablo had already begun its construction in 1567; therefore, that initial construction was a parish church in the Hanan Saya part of the town of Ácora. Likewise, Frisancho (1999) agrees with the investigation of Gutierrez and other authors, that the Church of San Pedro and San Pablo de Ácora had begun its construction in the middle of the year 1567, since Ácora, in those times, was considered to be one of the seven most important towns attached to the Chucuito district. In support of Meiklejohn's (1988) research on the churches and the lupacas during the colony, this indicates that, in 1553, the Dominicans did not have a church for the work of evangelization, and for this reason, they decided to promote the start of the construction of a church, the same one that occurred in the middle of the year 1560, since it later adopted the name of San Pedro and San Pablo de Ácora, claiming that this affirmation belongs to Dominico Fray Tomas del Castillo (Gutiérrez et al. 1978).

*Construction masters*

As is known, architectural constructions in the colonial era were directed by masters from Spain. In this regard, the construction of the Church of San Pedro and San Pablo de Ácora, was directed by the teacher Alarife Escobedo, who took charge of the masonry part; the labor was charged to the mitayo Indians and tributaries of the ayllus of Ácora. On the other hand, in the investigations of Gutiérrez et al. (1978, p. 284) it is said that:

> two thousand Indians were in the work and that the others who are in these works serve for their mitas . . . in turn the Indians of the Urin Saya de Ácora partiality declared that they should have made by order of the lawyer Estrada, a corregidor, 230 pieces of clothing and that the money they gave for it was given to Escobedo who does the work of the Church.

As for the carpentry work, the Spanish master Bustamante was commissioned, who regularly did the interior carvings of the church. However, the work was not free, since, if the church needed decorative ornaments, then Mr. Bustamante made the payments in kind or money (Gutiérrez et al. 1978).

*Completion of construction*

Like in any construction process, there is a culminating part, so the Church of San Pedro and San Pablo is no exception. In this regard, Julie and Julien and Toledo (1998), in their research, took as a reference the report of Pedro Valencia, who was the Bishop of La Paz and made his report in 1619, since in this, he indicated that, by 1579, in the town of Ácora, the Church of San Pedro and San Pablo was already finished. In this regard, Gutiérrez et al. (1978) confirmed that, in 1610, the church was already completely finished, data that were taken from the reports of Father Nicolas de Santa María. Likewise, the implementation of the interior components was carried out gradually in the following years; for this, there is information in the studies by Frisancho (1999), where he mentions that, in 1771, the priest Sebastián de la Riva had the main altarpiece and its images made and gilded them in 1776. Therefore, this information continues to be true and dominant in the 21st century.

*Reconstruction phase*

The Church of San Pedro and San Pablo maintained the first phase of its construction until the beginning of 1780, but, as a result of the indigenous uprisings in the Peruvian-Bolivian Altiplano against the Spanish rule led by Túpac Katari, it was truncated. It is worth mentioning that the successive riots attacked the Church of San Pedro and San Pablo, since it was set on fire and badly damaged. Thus, to rebuild the church, the parish priest José Erazo de Burunda was the one who faced the repairs, for which they ordered wood to be brought from Larecaya and put the new roof, whose masters came from Cusco and La Paz (Meiklejohn 1988). However, a short time later, the tower suffered a collapse due to natural factors (rain) and, again, it was rebuilt, so the temple is still standing.

In 1924, Doctor Juan M. Mariscal, with tenacious work, managed to change the roof with calamine and a large part of the nave. After this, the people had him as a benefactor of the temple. However, in the times of the previous century, successive renovations were carried out by the prelature of Cusco, which lasted until the end of the 20th century and, after that, the church was simply gradually abandoned by the authorities and population of Ácora. Probably in current times, faith and imposition, although continuing in some way, no longer have hegemony or much impact.

*Current conditions of the churches*

The referential degree indicates that the Church of San Pedro and San Pablo is located two blocks from the main square of the District of Ácora, between the intersection of the Jirón Libertad and San Pedro streets. That is to say, the architectural construction of the Church of San Pedro and San Pablo de Ácora has a validity that dates from the end of the 16th century, lasting four centuries; in the historical course of the Church, the reconstruction

phases that have made its infrastructural conservation possible up to the present day arise, which are described below.

*External features of the Church*

- Front facade: There are two clearly differentiated spaces, as shown in Figure 1: the central nave and the presbytery.
- Central nave: The main doorway is made up of the image front of the church that still preserves the characteristics and details that were made by the Dominican religious orders. On the sides of the main portico, there are two pilasters on each side; the first two extend to the junction of the second cornice and the two outer pilasters significantly exceed the triangular ogive.

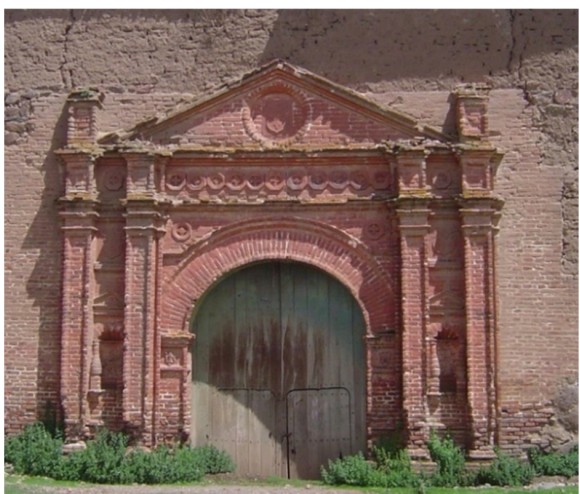

**Figure 1.** Church front image. Source: Photograph by the authors, August (2021).

Likewise, the four pilasters form three well-known entrecalls; the main arched portico is located in the middle aisle of the main portico; in the vertical middle of the portico, there are two human faces carved in brick in high relief and their features show signs of being a possible Spanish character. The outer lanes are narrow and, in the lower part, there are two niches (one in each lane); in the upper part of the portico, there is a triangular ogive that extends above the second cornice and, in the middle part, there is a recessed circle, which, for the Dominicans, was considered to be a medallion, since the shield is clearly observed of the Dominicans as a sign that affirms that the church was built during the permanence of such religious order. Therefore, the space of the main doorway is built with golden brick and joined with lime mortar, as shown in the following Figure 2.

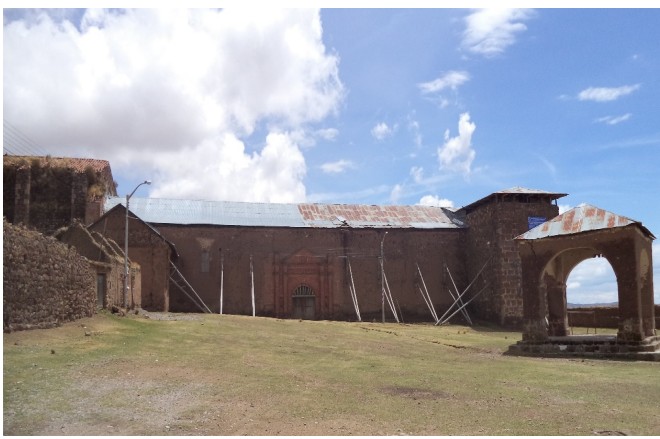

**Figure 2.** Church front image. Source: Photograph by the authors, August (2021).

Another component found in the central nave is the sacristy room that is on the right side of the imafront attached to the molding of the presbytery; on the left and right sides of the main portal, a window can be seen, that is, the window on the right side is notorious and has a recessed shape, while that on the left side is completely covered with adobe, stone, and mud. In this logic, it is indicated that, on both sides of the imafronte, there were walls supporting the wall, on the left side, two walls, and on the right, a wall that collapsed some sixty years ago due to the effects of rain. At present, it is observed that, to avoid the collapse of the wall, the wall is supported with props made of eucalyptus logs.

- **The presbytery:** This is located on the right side of the central nave built with stone with mud mortar; the construction space is clearly outstanding, externally, it has four buttress columns supporting the wall (four on each side), the same ones that form the three entrecaldes in the front part, and in the upper part, there is a recessed window. Likewise, in the space of the atrium, there is an open chapel, as shown in Figure 2, since this chapel had the function of sheltering the priest so that he could lead the holy mass with the concurrence of indigenous people of a large crowd.

- **Rear facade:** No detail can be seen in the part of the nave, there are only signs of support walls detached from the wall; as for the rear space, this is currently supported with eucalyptus beams. On the other hand, the presbytery has four support columns that form three entrecalls; in the outer passageways, there are two arched and recessed windows that are currently covered with stone and mud. Moreover, in the lower part of the middle passageway, there is a door (apsidial door) in the shape of an arch, the same one that is covered with stones and cement seen in Figure 3.

- **Right- and left-side facades:** The right facade belongs to the presbyter, since two buttress walls that stand out in the lower middle part are found in an arched door, the same one that is currently covered with adobe and mud. On the other hand, on the left facade, there is the second doorway of the church, which is also called the doorway of the feet, and its materials are the same as those of the front doorway, with the difference of it only having two recessed pilasters that form a entrecalle, ending in a closed triangular pediment. After these columns, there is a continuous cornice that forms a horizontal entrecalle, that is, under the entrecalle, there are some medallions with a center recessed by a small carved cross.

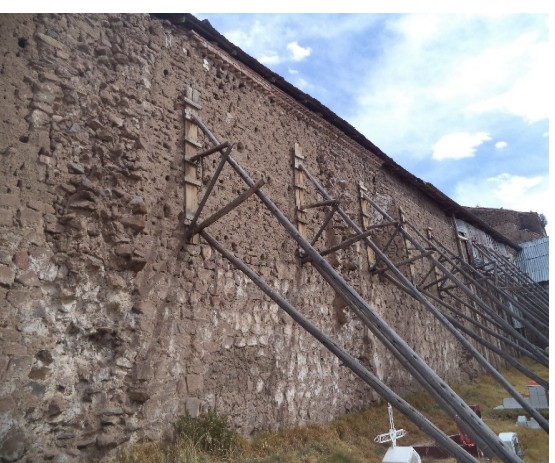 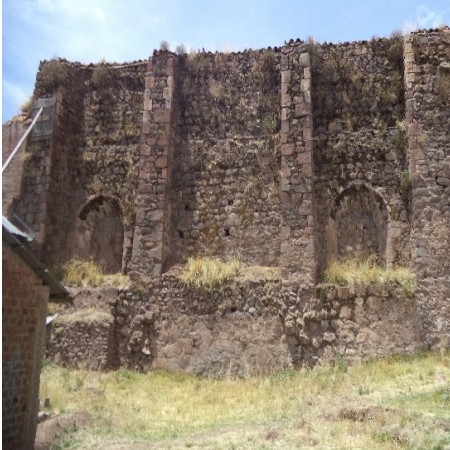

**Figure 3.** Rear facade of the Church of San Pedro and San Pablo. Fountain: Photograph by the authors, August (2021).

Finally, there is a triangle whose interior does not have any image or carving, but, on the contrary, on the triangle there is a rectangular window, which, today, is covered from the inside with calamine; In addition, in the upper part of the window, there is a circular shape made of brick and lime. The details of the side covers are shown in Figure 4.

**Left-side facade**          **Right-side facade**

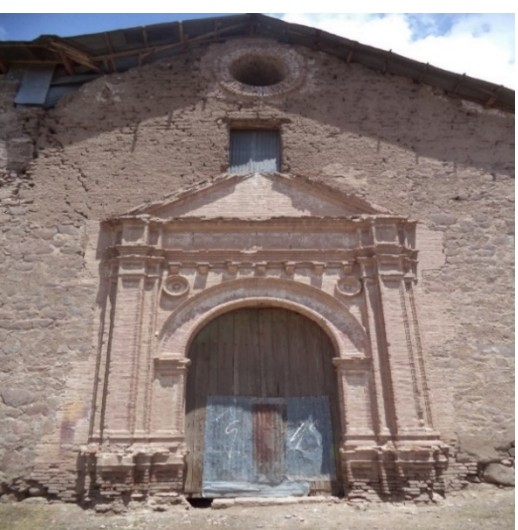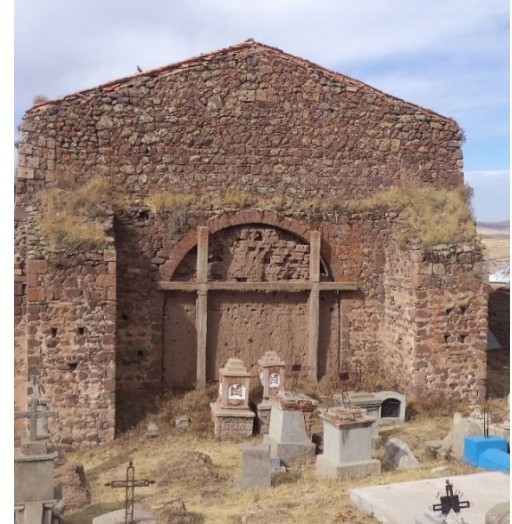

**Figure 4.** Rear facade of the Church of San Pedro and San Pablo. Source: Photograph by the authors, June (2021).

- **Roof of the church:** Part of the central nave is roofed with calamine and part of the presbytery still has a tile roof.
- **The church tower:** This is located on the left side of the foot portal, attached to the wall of the central nave with the body notably protruding. The materials used are purely stone and lime, but currently, only the base of the tower is notorious, and even the bell is found in the municipality of Ácora.

**Details or interior components of the church**

The internal implementation of the Church has gone through several processes and changes, because the wall is covered with plaster; likewise, at the junction of the nave with the presbytery, there is a large brick arch covered with plaster, and in the upper part of the arch, there are two small windows. It should be added that three protruding arches were found that arise from the outer buttresses of the wall, since there is a plaster carving of the apostles of Jesus Christ, which is bathed in gold leaf. On the other hand, in the upper part of the presbytery, there is a union with an extended roof to the outline of the protruding interior cornice, with details of high Andean plants Figure 5.

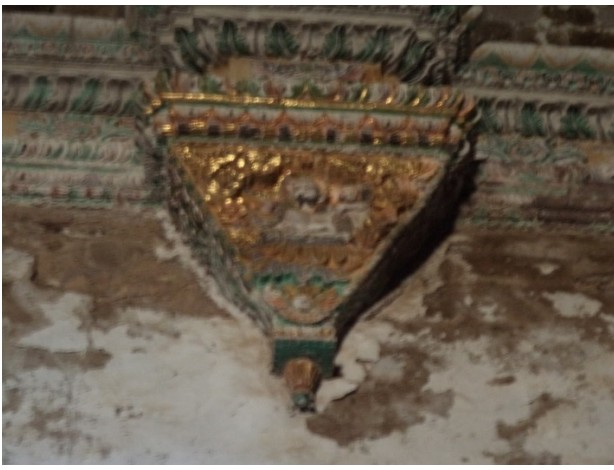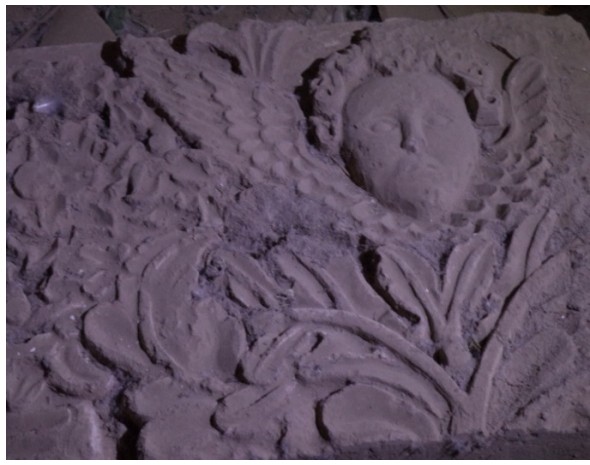

**Figure 5.** Interior details of the Church of Saint Peter and Saint Paul. Source: Photograph by the authors, June (2021).

### San Juan Evangelista Church
### Construction promoters

Taking into consideration the investigations of Meiklejohn (1988) and Gutiérrez et al. (1978), the promoter of the construction of the San Juan Evangelista Church was the parish priest Blas Moreno Hidalgo, who had the mission of building the church from its foundations, so that the said construction of the work began at the beginning of 1590, and its respective culmination was reached in 1620.

*Reconstruction phase*

Due to factors contrary to the Church, according to Gutiérrez et al. (1978), the temple was partially destroyed in 1696, and by the 18th century, it had to be rebuilt by the parish priest Jose de Estrada. Moreover, in 1711, the master mason Salvador, who was from the town of Ilave, carried out work on the arch of the cemetery, an image of which can be seen in Figure 6. Between 1724 and 1731, the faithful donated various elements to implement the temple, and bells were also placed in the tower, which were made by a French blacksmith. In 1740, they commissioned the gilding of the tabernacle and braziers, while the master organist Nicolas Laguna made a new instrument for the temple that was finally completed by the master Isidoro Muriel in 1751 (Parish Archive of Ácora 1740–1760).

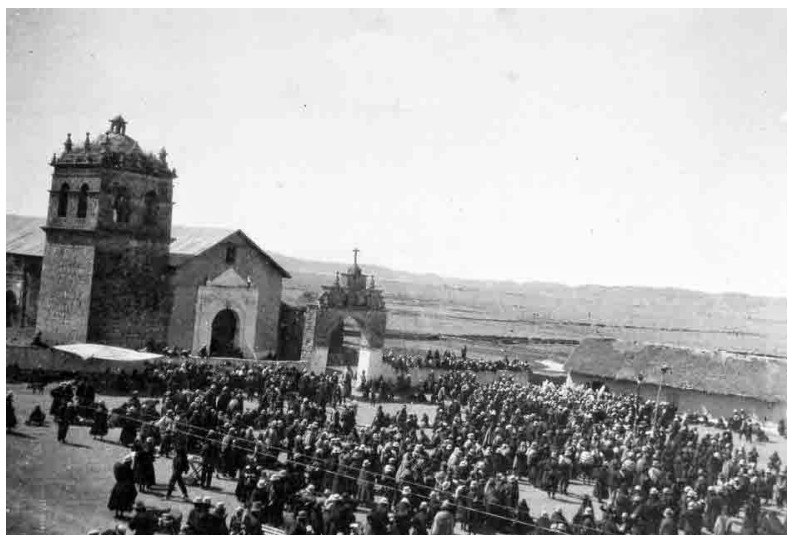

**Figure 6.** Arch of the cemetery Church of San Juan Evangelista. Source: District Municipality of Ácora. Reproduced with permission of the district alderman.

In 1753, the altarpiece and main tabernacle of San Juan was composed, and in 1757, a new altarpiece was made under the invocation of San Antonio, which was made by the master carpenter Pablo Santa María. Likewise, in 1776, a new stage of supplying wooden works for the churches of Ácora began, this time coming from Cochabamba. In 1785, the church was completely whitewashed and, in the choir part, stones of aubergine were placed. For this reason, Gutiérrez et al. (1978) maintained that the San Juan Evangelista temple remained in good condition during almost the entire 19th century; however, in 1887, due to a fire, the temple was left in ruins. It was in such a way that the then Bishop of Puno, Monsignor Puirredon, ordered its reconstruction for the year 1891. Consequently, stribus were placed on the walls and a calamine cover was placed early in the Collao, which was carried out on 16 October 1897, thanks to the contribution of the donations of Feliciano Guzman de Aguilar who, with praiseworthy intention, gave the necessary funds.

*Current conditions of the San Juan Evangelista Church*

The San Juan Evangelista Church in the Ácora district is one of its main monuments today, since its maintenance and operation have given some hope, especially to the Catholic faithful who celebrate mass in honor of different festivities and celebrations. The church is

located in the main square of the Ácora district and is perpendicular to the square (north to south), between the intersection of the jirón Arica and jirón Ácora streets.

*External features of the church*

The church occupies the main block of the district, and the main portal is perpendicular to the short side of the large rectangle facing the square, since this is the model that was used in the urban architecture of the Ácora district. On the other hand, the main doorway of the Church of San Juan Evangelista does not have many details, and there are only two quadrangular pilasters that extend to the side of the portico, to the junction with the cornice; in the middle part, there is the main portico entrance to the church, which is arched at the beginning of the arch, and two notoriously protruding capitals can be seen. Meanwhile, in the upper part of the cornice and at the height of the pilaster, there are two small caves, since the inner part extends to the frontal triangle carved in two moldings and the middle part is flat without any detail; also, at the top of the triangle, there is a rectangular window.

As for the tower, this is located on the right side of the image front, divided into two mouldings; the first one is in the entrance door to the bell tower and, at the same time, a small recessed window fulfills the function of giving light to the interior; between the first and second molding is the capital divided into three stairways, on the upper part the second molding is borne, and in it are the eight arches of the bell tower Figure 7. In addition, in the upper part with the union of the dome, it is bordered by a strip of capital and, on top of it, the oval-shaped dome is borne; in the upper part of the strip of the capital, there are small pillories of various shapes and, in the upper part of the dome, there is a small open chapel with a cross.

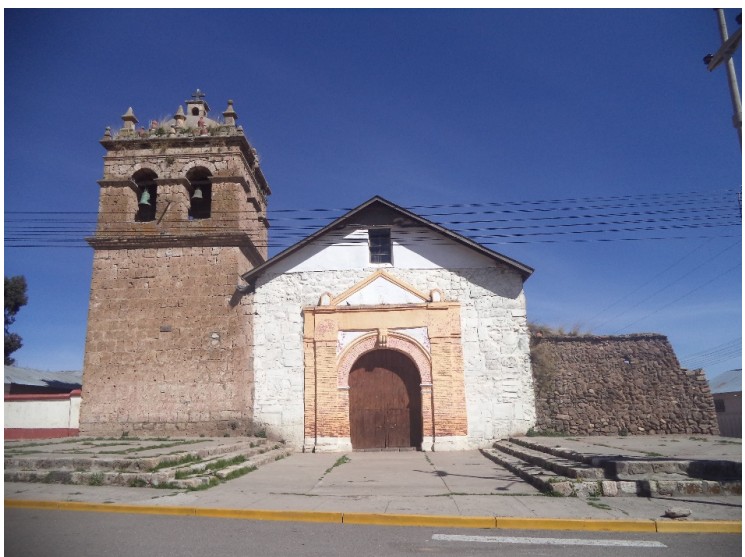

**Figure 7.** Front facade of the Church of San Juan Evangelista. Source: Photograph by the authors, June (2021).

Based on the previous figure, the imafronte is also located in the central part of the nave, considering that the side portico that lacks detail is located there; moreover, on the outer sides, there are two pilasters that extend to the height of the second cornice; in addition, in the upper part, there is a frontal triangle and, in the middle, there is a small sunken grotto. Finally, the entire space of the portico is made of brick material with lime mortar. However, at present, this space is in a state of deterioration, which is why some parts of the portico have been refurbished with sand and cement material Figure 8.

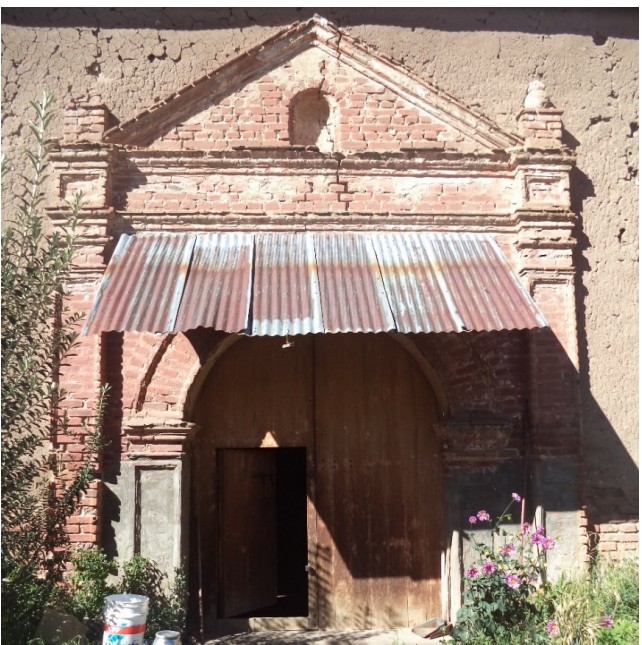

**Figure 8.** Side facade of the Church of San Juan Evangelista. Fountain: Photograph by the authors, August (2021).

**La Concepción Church (La Concebida)**
*Start of construction and developers*

Following the chronology of Gutiérrez et al. (1978), the beginning of the construction of La Concebida Church dates back to the end of the 16th century, that is, in 1595. At that time, the San Juan Evangelista Church was already under construction. According to the investigations of the authors already mentioned, the names of those who started the construction are not available. If this can serve as an indication, in 1634, Bishop Feliciano de la Paz indicated to the priest Diego de Tapia y Montalvo that the doors to the temple were missing and the walls were mistreated, so they should be whitewashed. Likewise, in those years, it was mentioned that some openings would be closed with adobe "until there are wooden doors to put them in and they do the works so that it is with the decency and reverence that is due to the temple of God and his house" (Parish Archive of Ácora 1630–1687).

*Teachers*

With respect to the builders of the construction, no names have been found, but as for those who were ordered to make the ornaments, there are some names, such as: tailor Diego Gonzales, silk Diego de García, and foundry master of bells Francisco de Medina. In addition, other masters were involved in the reconstruction of the church, and among them were: master organ builder Juan Crisóstomo, who was given an advance of two hundred pesos for the elaboration of the organ, and foundry master Diego Felipe, who was in charge of melting the bells, both of whom were teachers and indigenous people of the town of Ácora.

*Culmination*

By the beginning of the year 1608, the church was finishing its construction, so two years later, it was found that the construction of the church had already been completed and that the necessary materials were implemented inside Figure 9.

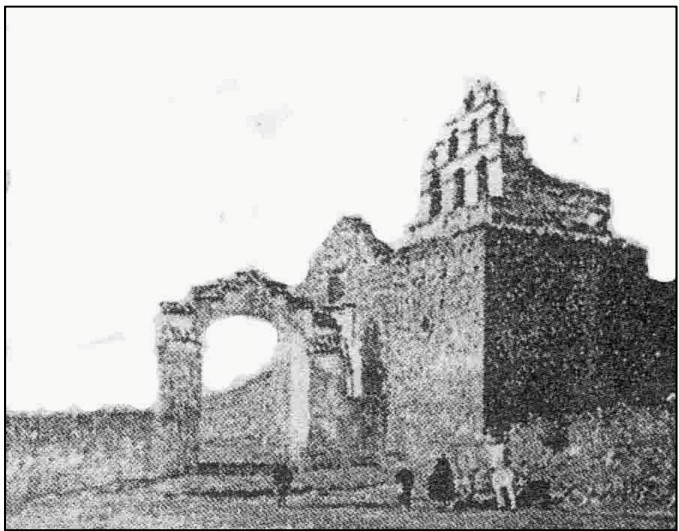

**Figure 9.** Construction of the La Concepción Church. **Source:** Parish Archive of La Paz (1900–1950). Reproduced with permission of the district alderman.

*Reconstruction phase*

The La Inmaculada Concepción or La Concebida Church was rebuilt at the beginning of the year 1700 and completed in the year 1710. Information found in a 1710 inventory of the Parish Archive of Ácora mentions that the new church had designs and reconstruction materials of wood, nailing, four stone and lime arches with their better haves, and three new doors in the Church of San Juan, with three tribunes that served as a choir, a new three-section altarpiece with a tabernacle, eight canvases, and an image throne with twelve small mirrors (Parish Archive of Ácora 1690–1720). After the repairs of La Concebida Church began, due to it being very deteriorated, they were completed in 1764, so the painting of the half orange and main altarpiece continued, making new bars in the presbytery. A decade later, the work of gilding the altarpiece was carried out, where "gold leaf" was brought from Cuzco and, in 1787, the other altarpiece was polished (Gutiérrez et al. 1978).

**Actual conditions**

La Concepción Church is located between the streets of jirón Arequipa and jirón Grau, with the interjection of jirón Junín. One of the residents of the Ácora district, Mr. Alfredo Cruz, commented that: "when I was a child, La Concepción Church still had its walls and roof, which were made of calamine . . . it also kept its tower, which was half made of stone and the other. another one made of adobe . . . but due to the lack of maintenance by the authorities, there is nothing left of its construction Figure 10". On the other hand, Mr. Mendoza, 78, commented that, in his childhood, he only saw the walls of the church, and as an adult, the church was about to collapse, so the Ácora authorities themselves had to tear it down.

Until 1941, a large part of the structure of the arches and stone vault in the church, the arches in the cemetery, and the tower remained, but a decade later, they were dropped. At present, almost nothing of La Concepción Church's architecture remains, only the walled atrium that serves as a cemetery. As is to be expected, La Concepción Church was destroyed due to the insensitivity of the authorities and a change in the belief of the indigenous people, that is, every day, there are fewer followers of the Catholic religion.

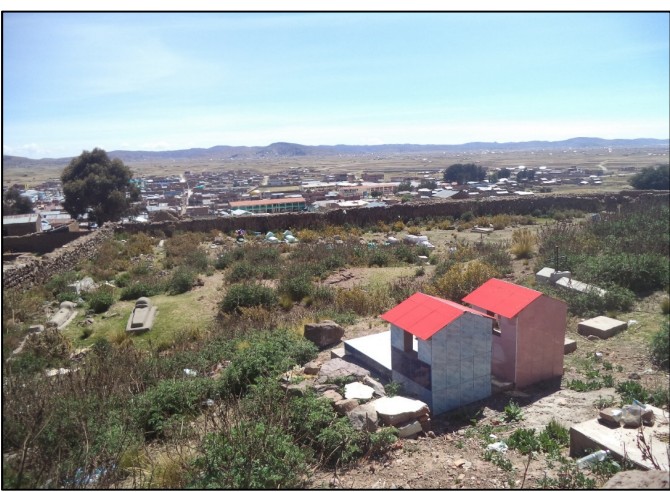

**Figure 10.** La Concepción Church Cemetery. Source: Photograph by the authors, August (2021).

**Santa Bárbara Church**

José Portugal Catacora, cited in Flores (2012), indicated that the parish priests of San Pedro and San Pablo, Pedro de Macoagua y Contreras, born in La Paz in 1670, stated the following regarding the church: "At the ends of that town he also had a Church built at the Glorious Santa Bárbara and that a sovereign relic was obtained for her". Consequently, the Santa Bárbara Church, according to Mr. Alfredo Cruz, had some walls and a tower that were built of adobe. Likewise, on November 29 and 30 of each year, some devotees performed a mass in honor of the patron saint Santa Bárbara; now, such mass is celebrated in the San Juan Church that is still standing. Thus, if the Santa Bárbara Church had been preserved, its location would be at the ends of the west side of the town of Ácora, having, as its main reference, the San Pedro square next to the jiron San Pedro, where a one-meter high wall still stands Figure 11.

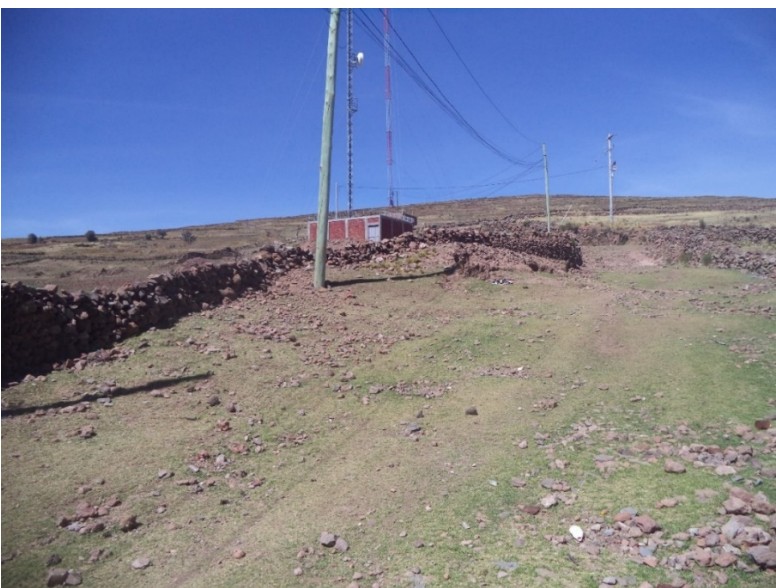

**Figure 11.** Place where the Santa Bárbara Church was located. Source: Photograph by the authors June (2020).

## Functions of the Churches of Ácora

### Functions of churches in colonial times

In the highlands of Puno and in Chucuito, Christianization began a decade after the Spanish conquered the area. Therefore, in colonial times, churches were very necessary for the first missions of the Dominican religious order, since they sent two missionaries to the main capitals in the province of Chucuito to fulfill the great mission of evangelizing the natives. On the other hand, the main function that the great buildings have fulfilled is gathering the apparent deviants or non-evangelized, since what was imparted from the clergy was one of the most effective weapons that would help to eradicate idolatry in the highland towns. However, in 1553, Dominican missionaries lamented that they did not have churches in other parts of the highlands, so it was essential to promote the construction of such churches and thus feed the Christian liturgy.

Another relevant aspect was the language, knowing that evangelization was carried out in the native language (Aymara). In the year 1684, the bishop of La Paz Monsignor Queipo Llano wrote a letter to the Spanish crown, in which he informed the King that, during his first visit to the province of Chucuito, he reminded the priests that they should teach Christian doctrine not only during Lent, and that they should do so in the native language (Meiklejohn 1988). On his second visit, which dates back to 1687, Monsignor Queipo Llano revealed that the parish priest of the town of Ácora, Fadrique Sarmiento, was not fulfilling his role of preaching and evangelizing, since he only did it in times of Lent, it was not him who directed, but his assistants, and he had not yet learned the native Aymara language. As is clear, evangelization in the native language was very important for the Spaniards, so the function of the priest was not only for people to know and repeat prayer, but also to explain the various points of Christian doctrine in their language, obligatorily (Archivo de La Paz 1684). On his third visit, the parish priest of the town of Ácora had already learned the native language and he faithfully preached on Sundays and holidays.

After the warning received, evangelization began in Ácora, as mentioned by Meiklejohn (1988). It is indicated that the friars gathered the children daily, while the adults met three times a week (Wednesdays, Fridays, and Saturdays) to recite and learn Christian doctrine. They also gathered at each patronal festival, either before or after mass. Attendance at these meetings was mandatory, and if they did not attend, the indigenous people were punished.

### Tax function of the Church

Another of the functions that church had in the colonial period was to collect the tithe and tercio tributes that were later sent to the Spanish crown. As usual, "the highland clergy were supported by the crown, and the crown obtained tribute money collected from its own indigenous parishioners" (Meiklejohn 1988, p. 167). Likewise, in the highlands, the offerings were presented but known as obentions, as shown in the following Figure 12.

On the other hand, the tributes contributed by the indigenous people of the highlands to ecclesial doctrines are systematically presented. This displays a short excerpt from the source Figure 13.

It should be considered that the document presented above is a request where Captain Don Nicolás Romero y Junter, a resident of the town of Ácora, is denounced by an indigenous person for the embezzlement of the taxes contributed by themselves to the Church. However, the document apparently did not prosper, since Nicolás Romero requests his release, alleging that there was insufficient evidence to implicate him in such an act of embezzlement.

**Curates of Huancane and Chucuito**

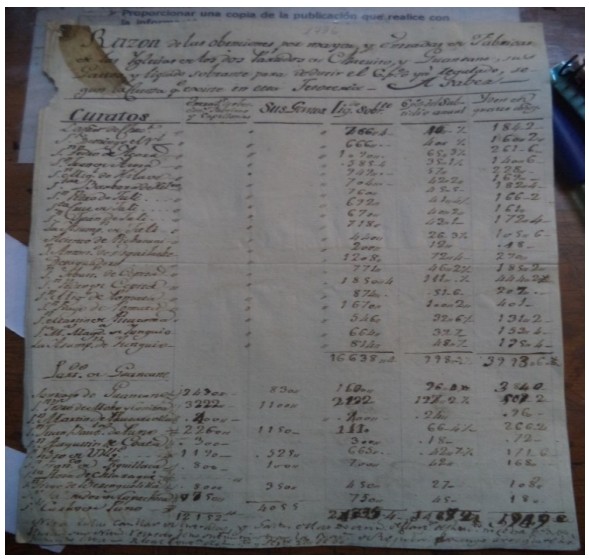

*"Reason for the wholesale and factory-sent gifts from the Churches of the two districts of Chucuito and Guancane, their expenses and excess liquid to deduct 6%, already regulated, according to the account that exists in this treasury–Namely".*

**Figure 12.** Document on the accounting of collection and expenses of the Church San Pedro and San Juan. **Source:** Historical Archive of Puno. Reproduced with permission of the district alderman.

**Original source**

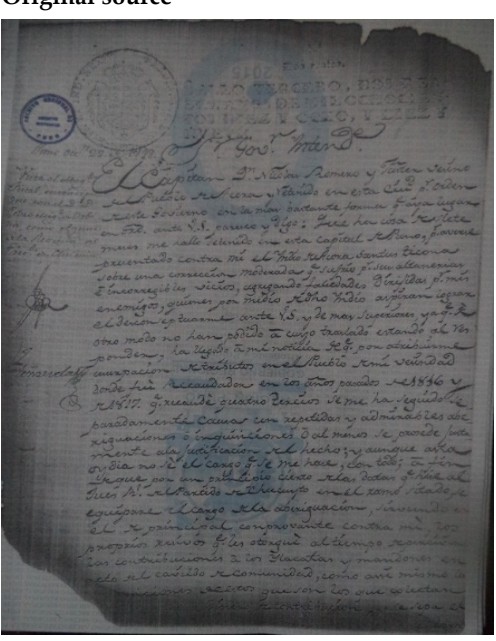

**Brief description**

*Proceedings*

*Mr Gov r Inten of*

*"The captain Mr. Nicolas Romero y Junter, a resident of the town of Ácora and detained in this city by order of this government in the most adequate manner that is possible... I look and say: that about seven months ago I have been detained in this capital of Puno, for having presented against me by an Indian from Acora Santus Ticona...and eight Indians... he has received my notice that for attributing to me the theft of tributes they gave the town of my vicinity where I was a collector in the past years 1816 and 1817 that I collected four thirds..."*

*"and I have never seen shortages, militias and hunger due to my misfortune in eight years… I caused the emigration of a large number of poor Indians… unable to make their payments…*

*Nicholas Romero"*

**Figure 13.** Management of taxes contributed by the natives and the doctrines. Source: Historical archive of Puno, 1816–1817. Reproduced with permission of the district alderman.

*Evangelizing function of the Church*

The Catholic Church has a considerable incidence in scenarios where it has put the cross as a symbol and God as the father of all creation. It was probably the verb induced to the indigenous and vehicle of colonial evangelization that has had the most relevance. The colony was the best scenario for the implementation of the catechization process for the peoples of the highlands. From the high ecclesial commands, they imposed their doctrine, to such an extent that all the priests had to speak a native language of the place (Mendoza 2016). In this regard, the author's analysis, already referenced, may give an idea about what is intended to be explained, knowing that evangelization is:

the creed, the Our Father, the Hail Mary and other recitations, prayers and chants, which converts had to learn by heart, above all things. Instilling the Christian faith in the natives through words was the most direct method that the doctrineros applied throughout the colony. Clearly explaining the contradictions between good and evil, between the objectives that God and the Devil have, to save or condemn human souls, became a primary objective in the catechization process. They emphasized the idea of soul salvation towards eternal life, fulfilling the commandments dictated by God and not falling into sin, in a world where life is full of sinful actions, which must be avoided. The religious orders committed themselves with full conviction to offer the glory of God to those who accepted and assimilated to the Christian faith and do not live in sin, as well as to explain to them that non-Christians and sinners await them in the afterlife, the abominable hell ruled by Devil, representative of evil. For all this, converting the infidels to Christianity was a difficult and complicated task. (Mendoza 2016, p. 58)

Definitively, the fear implanted in the indigenous people worked perfectly, since, until the 21st century, the insistent glorification of God and the other forms of worship learned throughout the centuries persist (Estermann 2014). However, what was a matter of glorification and holiness has become a power game between the Catholic Church and the evangelical sects. From this perspective, Pérez Guadalupe (2017) pointed out that the Catholic Church has the advantage of a greater stability, history, tradition, an indisputable institutional hierarchy and unity, and a state, the Vatican, which has an experienced a diplomatic apparatus and long tradition of international negotiations. For their part, evangelical churches have a growing number of faithful, are more conservative and even pro-government, and have a more religiously established membership.

### 3. Conclusions

Once the files that explain the main function of the Catholic churches in the Peruvian highlands are shown, the Spanish domination over indigenous civilizations is exposed, because the real purpose of Europeans was to extirpate the idolatry of those who lived for thousands of years under guardianship of the cosmos. Thus, evangelization was carried out with contents of violence and imposition towards non-believing men and women. However, these events were not recorded in main regional archives, nor were they treated seriously.

On the other hand, the historical process of the churches of Ácora has a diachronic nomenclature based on the oldest ecclesiastical constructions established by the Spaniards in the space of the highlands of Puno, which began in 1560 with the construction of the Church of San Pedro and San Pablo; then, the churches of San Juan Evangelista, La Concepción, and La Santa Bárbara were built, until the sunset of some of them and their maintenance both by the authorities and some religious faithful who still find meaning in the Catholic Church.

The history of the Catholic churches in the Altiplano began from the moment of their construction, which was devised by the religious order of the Dominicans and later led by the secular clergy and Jesuits who, in addition, took charge of their maintenance, implementation, and reconstruction, exclusively for the purpose of paying attention to the Catholic parishioners who have remained until present time. Once arriving at their material chronology, it was observed that the main characteristics of the churches of San Pedro, San Pablo, San Juan Evangelista, and La Concepción manifest rectangular shapes and are of an ample size, since their design is a Latin cross plan. Instead, the Santa Bárbara Church presents a small quadrangular shape.

### 4. Materials and Methods

The methodological design was based on qualitative research and was a documentary review with a phenomenological, interpretative nature (Huanca-Arohuanca 2021). Likewise, the ecclesial monuments of Ácora were contrasted in situ and with a regular

frequency in the temporality from the years of 2016 to 2021. Therefore, their analyses and descriptions give a more complete idea of what they were in the colony and what they are at present.

*Information treatment plan*

The process and collection of the information was systematic; therefore, it carried out the following activities:

○   First: information was obtained through a literature review and from primary source documents. In this process, the method used was hermeneutics, so that the information collected was recorded in the guides and bibliographic analysis sheets, and was later recorded in the construction of the manuscript.

○   Second: then, the information was collected in situ through participant observation. Likewise, the information was recorded in observation form and, at the time of the data collection, a photographic camera was used to capture the evidence.

In order to end the research in a consistent manner, it was written according to the quality standards required by the scientific community. The fact of following what has been mentioned took longer than expected, affecting the dissemination and subsequent reproduction of the information.

**Author Contributions:** Conceptualization, J.W.H.-A. and E.G.-G.; methodology, G.C.-R.; software, F.S.-C.; validation, J.W.H.-A., E.G.-G. and D.M.-J.; formal analysis, J.W.H.-A.; investigation, E.G.-G.; resources, G.C.-R.; data curation, J.W.H.-A.; writing—original draft preparation, E.G.-G.; writing—review and editing, D.M.-J.; visualization, F.S.-C.; supervision, F.S.-C.; project administration, D.M.-J.; funding acquisition, E.G.-G. All authors have read and agreed to the published version of the manuscript.

**Funding:** This research received no external funding.

**Institutional Review Board Statement:** Not applicable.

**Informed Consent Statement:** Not applicable.

**Data Availability Statement:** No new data were created or analyzed in this study. Data sharing is not applicable to this article.

**Conflicts of Interest:** The authors declare no conflict of interest.

## References

### Archives and Primary Sources

Archivo Histórico de Ácora [Ácora Parish Archive]. (1740–1760). *Eclesiásticos, páginas sueltas* [Ecclesiastics, loose pages].
Archivo Histórico de Ácora (1690–1720). *Libro de fábrica de la iglesia Concepción* [Factory book of the Concepción Church].
Archivo Histórico de Ácora (1630–1687). *Libro de fábrica de la iglesia Concepción* [Factory book of the Concepción Church].

### Secondary Sources

Archivo de La Paz. 1684. *Ordenanza de Celebración de Misa en las Doctrinas*. Available online: https://archivolapaz.com.bo/ (accessed on 11 August 2019).
Barria-Asenjo, Nicol A., Slavoj Žižek, Hernán Scholten, David Pavón-Cuellar, Gonzalo Salas, Oscar Ariel Cabeza, Jesús William Huanca Arohuanca, and Sergio J. Aguilar Alcalá. 2022. Returning to the Past to Rethink Socio-Political Antagonisms: Mapping Today's Situation in Regards to Popular Insurrections. *CLCWeb: Comparative Literature and Culture* 24: 1–13. [CrossRef]
Castillo, L. J. 2019. *Templos Barrocos del Collao*. San Borja: Ministerio de Cultura.
Espinosa, Oscar. 2018. La relación de la Iglesia católica y las Iglesias evangélicas con las organizaciones indígenas en la Amazonía peruana: La experiencia del pueblo achuar. *Bulletin de l'Institut Français d'études Andines* 47: 267–92. [CrossRef]
Esquivel, Juan Cruz. 2000. Iglesia Católica, Política y Sociedad: Un Estudio de las Relaciones Entre la elite Eclesiástica Argentina, el Estado y la Sociedad en Perspectiva Histórica. Available online: http://biblioteca.clacso.edu.ar/clacso/becas/20110112035544/esquivel.pdf (accessed on 11 August 2019).
Estermann, Josef. 2014. *Cruz y Coca. Hacia la Descolonización de Religión y Teología*. Quito: Ediciones Abya-Yala.
Fazio, Hugo, and Daniela Fazio. 2018. El tiempo y el presente en la historia global y su época. *Revista de Estudios Sociales* 65: 12–21. [CrossRef]
Flores, L. 2012. *Historia del Distrito de Ácora*. Puno: Sangre Aymara.

Frisancho, I. 1999. *La Catedral de Puno*. Puno: HOZLO.

González, María del Carmen Díez. 2009. Aproximación a la arquitectura de los templos cristianos y musulmanes desde sus orígenes hasta finales del siglo XV. Semejanzas, diferencias y evolución. *Cauriensia* 4: 191–213. Available online: https://dialnet.unirioja.es/servlet/articulo?codigo=3086405 (accessed on 11 August 2019).

Gutiérrez, Ramón, Carlos Pernaut, Graciela Viñuelas, Hernán Rodríguez, Elizabeth Kuon, Estela Benavides, and Jesús Lambarri. 1978. *Arquitectura del Altiplano Peruano*. Puno: Libros de Hispanoamérica.

Huanca-Arohuanca, Jesús Wiliam. 2021. Narrativas de guerra y resistencia: Participación de la mujer austral del Perú en la Guerra del Pacífico. *Encuentros. Revista de Ciencias Humanas, Teoría Social y Pensamiento Crítico* 13: 50–59. [CrossRef]

Huanca-Arohuanca, Jesús Wiliam, and Néstor Pilco Contreras. 2021. Transición del virreinato a la República: Caleidoscopio sociopolítico-económico del altiplano puneño en la Independencia de Perú. 1815–1825. *Diálogo Andino* 65: 379–91. [CrossRef]

Julien, Catherine J., and Francisco de Toledo. 1998. *Toledo y los Lupacas*. Lima: Carlos Milla Batres.

Meiklejohn, Norman. 1988. *La Iglesia y los Lupacas Durante la Colonia*. Cusco: Instituto de Estudios Aymaras.

Mendoza, Román Robles. 2016. Arquitectura religiosa en los Andes: Apogeo, crisis y restauración. *Investigaciones Sociales* 20: 53–68. [CrossRef]

Molina Fuentes, Mariana Guadalupe. 2012. La Iglesia católica en el espacio público: Un proceso de continua adecuación. *Política y Cultura* 38: 49–65. Available online: https://www.scielo.org.mx/pdf/polcul/n38/n38a4.pdf (accessed on 15 August 2019).

Montaño, Eugenia Allier. 2018. Balance de la historia del tiempo presente. Creación y consolidación de un campo historiográfico. *Revista de Estudios Sociales* 65: 100–12. [CrossRef]

Paredes Cisneros, Santiago. 2018. Iglesias en Tierradentro: Edificación, uso y sentido entre los indios páez de la Gobernación de Popayán, siglos XVII-XVIII. *Revista de Estudios Sociales* 64: 55–74. [CrossRef]

Pérez Guadalupe, José Luis. 2017. *Entre Dios y el César. El Impacto Político de los Evangélicos en el Perú y América Latina*. Lima: Instituto de Estudios Social Cristianos (IESC).

Sausa, Mariella. 2018. El 76% de peruanos es católico, pero solo el 10% es fiel a la Iglesia. *Perú*. pp. 1–10. Available online: https://peru21.pe/peru/papa-francisco-peru-76-peruanos-catolico-10-fiel-iglesia-391759-noticia/ (accessed on 15 August 2019).

Topolski, Jerzy. 1973. *Metodología de la Historia*. Madrid: Cátedra.

Vargas, Rubén. 1966. *Historia general del Perú: Virreynato 1551 a 1596. Tomo II*. Lima: Carlos Milla Batres.

