# Peer review of "Catholic Churches of the Colonial Period in the Southern Andes of Peru: An Evocation towards the Past"

_religions, doi:10.3390/rel14070940_

Round 1

Reviewer 1 Report

Overall, the paper has some points that need significant work and development. First, it is unclear the scope/purpose of the paper. Perhaps a better constructed conclusion can bring all of the ideas into better clarity, allowing for a better understanding of what the paper achieved. There is also not a good integration of explaining how the arch. explored is connected to some of the claims on the violence and engagement of colonialism. There are also several differed claims that are not supported nor fully engaged in as far as they neglect supporting documents from the Catholic Church, etc. - it needs to be better explained in relation to what is "now" and what is "then". - The author is commended on engaging this topic, but they should look back to Church documents as a guide to outline what they are interpreting in conversation with what they are observing (especially those of the 14th-17th century that can explain better the Church's understanding of evangelization at that time). 

Some claims are not supported. Pleas see attached documents for comments. 

There are some awkward sentences and other language issues that need to be addressed. Please have a good editor attend to paper. 

Author Response

Thank you for the comments and suggestions noted in the manuscript, the suggestions have been corrected as requested. The English language was reviewed again.

Reviewer 2 Report

The article fulfils the formal and substantive criteria for scientific articles. The author has clearly defined the aim and method of the research work. The evaluations formulated are balanced and well-grounded. Despite the painful history associated with the sixteenth-century evangelisation of the indigenous peoples of the Andes, the author does not simplify the story, but draws attention to the complexity of the various aspects.  An additional value of the article is the photographic documentation of the churches, which are a symbol of a painful past but at the same time a valuable contemporary architectural monument. 

Author Response

Thank you for your comments and suggestions on the referenced manuscript. We are flattered by your appreciation. 

Reviewer 3 Report

See the attachment

No comments

Author Response

Thank you for the comment on the submitted manuscript, we have corrected the remarks according to the suggestions. In the resubmission of the manuscript, the English language was revised again.
